# Algo-Functional Indexes and Spatiotemporal Parameters of Gait after Sacroiliac Joint Arthrodesis

**DOI:** 10.3390/jcm9092860

**Published:** 2020-09-04

**Authors:** Chiara Busso, Simone Cambursano, Alessandro Aprato, Cristina Destefanis, Agnese Gianotti, Giuseppe Massazza, Alessandro Massè, Marco Alessandro Minetto

**Affiliations:** 1Division of Physical Medicine and Rehabilitation, Department of Surgical Sciences, University of Turin, 10124 Turin, Italy; chiara.busso@unito.it (C.B.); cristina.destefanis@yahoo.it (C.D.); agnesegianotti@gmail.com (A.G.); giuseppe.massazza@unito.it (G.M.); 2Department of Orthopedics and Traumatology, C.T.O. Hospital, Città della Salute e della Scienza, 10126 Turin, Italy; simone.cambursano@edu.unito.it (S.C.); ale_aprato@hotmail.com (A.A.); alessandro.masse@unito.it (A.M.); 3Division of Physical Medicine and Rehabilitation, Presidio Sanitario San Camillo, Fondazione Opera San Camillo, 10131 Torino, Italy

**Keywords:** arthrodesis, gait analysis, iowa pelvic score, majeed pelvic score, oswestry disability index

## Abstract

Aims of the study were to evaluate the reliability and validity of the Italian version of the Majeed and Iowa questionnaires and to investigate the long-term surgical outcomes following sacroiliac joint arthrodesis. Twenty one patients who underwent a sacroiliac joint arthrodesis and 21 healthy subjects were evaluated. The experimental procedure consisted of gait analysis and a physical activity assessment (in both groups) and of administration of outcome questionnaires and pain assessment (in the patient group). The Majeed and Iowa questionnaires showed excellent reliability, excellent (for the Majeed questionnaire) and good (for the Iowa questionnaire) construct validity, and poor convergent validity (for both questionnaires) relative to walking speed. Most of the patients reported no pain and minimum pain-related disability and their physical activity profile was comparable to healthy controls. Patients showed an impaired walking performance (i.e., they walked slower and using shorter steps) compared with healthy controls. Long-term walking pattern abnormalities following sacroiliac joint arthrodesis may occur despite excellent clinical results. Given their excellent reliability and construct validity, the Majeed and Iowa questionnaires can be used in combination with the assessment of spatiotemporal gait parameters for the prognostic assessment and/or follow-up of surgical patients.

## 1. Introduction

Sacroiliac joint fusion, also referred to as arthrodesis, is a surgical procedure that fuses the iliac bone (pelvis) to the sacrum (spine). It is performed for a variety of orthopedic conditions including fractures and spinal instability.

Sacroiliac joint fusion may be performed as a minimally invasive procedure or as an open surgical procedure requiring relatively large incisions, significant bone harvesting, and lengthy hospital stays [1]. The recent trend in orthopedic surgery has been to explore minimally invasive approaches such as the percutaneous sacroiliac joint fusion in which threaded cages or iliosacral screws, with or without bone graft, are placed percutaneously in order to achieve a fusion [1,2].

Despite various treatments, patients with pelvic injuries still have a spectrum of poor to excellent outcomes [3,4,5,6,7] that are assessed in both clinical practice and research through pelvic-specific outcome instruments [8,9] such as the Majeed pelvic score [10] and the Iowa pelvic score [11]. To our knowledge, no Italian version of these condition-specific patient-reported outcome measurement tools is currently available. Moreover, we are not aware of previous studies investigating the surgical outcomes of pelvic injuries in term of objective function such as the spatiotemporal parameters of gait (that represent a useful tool to assess the lower limb function in orthopedic patients) [12,13]. For example, it has been previously demonstrated that walking pattern abnormalities can occur after total knee [14] and hip [13] arthroplasty despite excellent clinical and radiographic scores and can last for several years after surgery. Therefore, it may be hypothesized that walking pattern abnormalities can occur also after post-traumatic sacroiliac joint fusion given that pelvic injuries produce not only bone fractures but also damage to soft tissues including the hip abductor muscles that play a crucial role during the stance phase of gait [13]. Thus, the aims of the present study were to: (i) evaluate the reliability and validity of the Italian version of the Majeed and Iowa questionnaires; (ii) investigate the long-term surgical outcomes following sacroiliac joint arthrodesis through a multidimensional patient evaluation including the assessments of pain intensity, algo-functional indexes, physical activity, and walking performance.

## 2. Materials and Methods

### 2.1. Patients and Study Design

Patients were randomly selected from postoperative lists and medical files of the “Centro Traumatologico Ortopedico (C.T.O.)” hospital (Turin, Italy). Twenty one patients (5 women, median (1st–3rd quartile) age: 49.0 (41.0–60.0) years; body mass index: 26.9 (26.0–29.4) kg/m^2^) who underwent a sacroiliac joint arthrodesis after fracture were identified and volunteered to participate in the study. The surgical procedure consisted of either open reduction internal fixation (performed in 9 patients) or percutaneous iliosacral screw fixation (performed in 12 patients). Post-operative assessment of fracture reduction was performed on the basis of the maximal displacement measured on the anteroposterior, inlet, and outlet pelvic views according to the following previously described criteria: excellent (≤4 mm), good (4–10 mm), fair (10–20 mm), and poor (>20 mm) [15]. In our series we had 12 excellent, 4 good, and 5 fair reductions. The present cross-sectional study was conducted after a median of 7 (min-max range: 5–10) years from surgery.

A control group of 21 healthy subjects matched for gender, age, and body mass index (5 women, median (1st–3rd quartile) age: 49.0 (42.0–58.0) years; body mass index: 23.6 (21.8–26.0) kg/m^2^) was also tested.

All subjects received a detailed explanation of the study and gave written informed consent prior to participation. The study conformed with the guidelines in the Declaration of Helsinki and was approved by the University of Turin ethics committee (protocol n. 133282).

### 2.2. Experimental Procedure

The experimental procedure consisted of gait analysis (see below) and physical activity assessment (through the Italian short version of the International Physical Activity Questionnaire-IPAQ short) [16]) in both groups of subjects. Moreover, the administration of outcome questionnaires and pain assessment were also performed in the patient group.

The original English versions of the Majeed and Iowa questionnaires were first cross-culturally adapted so as to be used for evaluating Italian-speaking subjects (see below). According to the COSMIN checklist [17,18], reliability (test-retest reliability and measurement error), validity (construct validity and convergent validity), and floor and ceiling effects (i.e., markers of responsiveness) of the Majeed and Iowa questionnaires were evaluated. All patients were asked to fill in the questionnaires twice (median number of days between the 1st and 2nd administration: 7 days) to evaluate their reliability. This reliability study design was chosen to prevent recall and to ensure that no clinical changes occurred between the two evaluation sessions. The construct validity of both questionnaires was evaluated by comparing their score (1st administration) with the Italian version of the Oswestry Disability Index (previously cross-culturally adapted and validated) [19], which was concurrently administered.

### 2.3. Cross-Cultural Adaptation

The cross-cultural adaptation process of the Majeed and Iowa questionnaires was performed according to previously published guidelines (e.g., the translators worked independently from each other, the items were translated forward and backward, translations were reviewed by bilingual people) [17,20] and comprised the following five steps. Step 1 included forward translation from English to Italian by two independent bilingual translators. Step 2 comprised the review of the versions produced by the two translators by a group of bilingual individuals, ensuring that the translation was acceptable to monolingual people, and their synthesis into one version. In step 3, the latter version of the questionnaires was translated from Italian back to English (back translation) by two independent bilingual translators. Step 4 comprised a consensus meeting of all individuals involved in the translation to review the back translation and decide on the final versions. Step 5 involved testing the final versions (Appendix A) in 10 consecutive subjects to examine the accuracy of wording and ease of understanding.

### 2.4. Self-Reported Questionnaires

The IPAQ short comprises seven items investigating different physical activity intensities (vigorous or moderate), the time spent walking and sitting (as a proxy for sedentary behavior) during the last 7 days [21]. Based on IPAQ results, three levels of physical activity were proposed in a categorical score: (1) low physical activity level (sedentary subjects)—IPAQ score below 600 MET*min/week; (2) moderate physical activity level (moderately active subjects)—IPAQ score above 600 MET*min/week and below 3000 MET*min/week; (3) high physical activity level (active subjects)—IPAQ score of at least 3000 MET*min/week.

The Majeed pelvic score [10] comprises seven items divided into the following five subscales: pain (30 points), work (20 points), sitting (10 points), standing (36 points total; walking aids, 12 points; gait unaided, 12 points; walking distance, 12 points), and sexual intercourse (4 points). The first category of each item was scored 0. Majeed suggested cutoffs for excellent, good, fair, and poor results in those working before the injury (85–100, 70–84, 55–69, <55) and those not working before the injury (70–100, 55–69, 45–54, <45).

The Iowa pelvic score [11] comprises twenty-five items divided into the following six subscales: activities of daily living (20 points), work history (20 points), pain (25 points), limp (20 points), cosmesis (5 points), and visual pain line (10 points) with 0 corresponding to “no pain” and 10 corresponding to “unbearable pain”. The original authors did not propose a grading scale.

The Oswestry Disability Index questionnaire, that examines the perceived level of disability in ten everyday activities of daily living [19,22], was adopted to assess the pain-related disability, as follows: minimum disability (0–20%); moderate disability (20–40%); severe disability (41–60%); crippling disability (61–80%); complete disability (81–100%).

### 2.5. Pain Intensity Assessment

Patients were asked to rate their pelvic pain intensity using a 11-point numerical rating scale (NRS), with 0 corresponding to “no pain” and 10 corresponding to “the worst imaginable pain”. Resting pain intensity was assessed prior to any study procedures, while movement pain was assessed during walking. Pain intensity was classified as mild for NRS scores between 1 and 3, moderate for scores between 4 and 6, and severe for scores between 7 and 10.

### 2.6. Gait Analysis

Spatiotemporal parameters of gait were measured in both groups with the use of the OptoGait photoelectric cell system (Microgate, Bolzano, Italy), which has been shown to provide reliable and valid data [23,24,25]. The OptoGait system used in this study consisted of 7 transmitting-receiving bars placed parallel to one another with 1 m between them (7 m × 1 m). Ninety-six light-emitting or light-receiving diodes are positioned on each bar 1 cm apart and 3 mm above the floor level. Data were sampled at 1000 Hz, processed using the OptoGait software program (version 1.12.15.0, Microgate, Bolzano, Italy), and stored for further analysis.

Participants, who were barefoot for the experiment, were asked to walk at two different speeds: self-selected comfortable (“walk at a pace that is comfortable for you”) and fast (“walk at a pace that is faster than you would normally walk”). At each speed, one familiarization trial always preceded three experimental trials and a rest interval of 60 s was observed between each trial. For each velocity, the following gait parameters were quantified as the average of the three walking trials: walking speed (m/s), cadence (steps/min), stride and step lengths (cm), stance and swing phase durations (ms), and double and single support durations (percentage of gait cycle). Moreover, the following other variables were compared concerning the operated limb of the patients, contralateral non-operated limb, and healthy subjects data (for the healthy subjects, left and right limb data were averaged): step length, stance and swing phase durations, and single support duration.

### 2.7. Statistical Analysis

The Shapiro-Wilk test for normal distribution of the data failed, and non-parametric statistical tests were therefore used. The Mann–Whitney test, the Fisher’s test, and the Kruskall-Wallis ANOVA (followed by Dunn’s post-hoc test, when the ANOVA was significant) were adopted for data comparison between healthy subjects and patients.

Changes in the questionnaire scores between the 1st and 2nd administration were analyzed with the Wilcoxon test to assess the presence of systematic bias.

Reliability (i.e., the degree to which the measurement is free from measurement error) of the Majeed and Iowa scores was assessed as test-retest reliability and measurement error [17,18].

Test-retest reliability (i.e., the extent to which scores from the same patients are unchanged for repeated measurements over time) was evaluated using the intra-class correlation coefficient (two-way mixed, single measure ICC2,1). A sample size of at least 20 patients was considered necessary for the test-retest reliability analysis, using the approximate method developed by Walter et al. [26] based on α = 0.05 and β = 0.20, indicating an expected level of reliability (ρ1) of 0.95 and a minimally acceptable level of reliability (ρ0) of 0.85. Power analysis was also performed according to previous studies [13,14] and showed that a minimum sample size of 18 subjects in each group was required to detect significant walking velocity differences between patients and controls (α = 0.05 and β = 0.05).

Measurement error (i.e., the systematic and random error of a patient’s score that is not attributed to true changes in the construct to be measured) was evaluated using: (i) the standard error of measurement (SEM) that was calculated as follows: √mean square error term from the ANOVA [27]; (ii) the smallest detectable change (SDC: i.e., the smallest individual change in a score that can be interpreted as a real change) that was calculated as follows: 1.96 × √2 × SEM [27].

The validity of the Majeed and Iowa questionnaires was assessed as construct validity (i.e., the degree to which a test measures what it is supposed to measure) by comparing their scores with the Oswestry Disability Index score and it was also assessed as convergent validity (i.e., the degree to which two measures that theoretically should be related to the construct do in fact correlate) by correlation analysis with walking speed, which is a well-established indicator of lower limb function in orthopedic patients [12,13].

Floor and ceiling effects of the questionnaires were considered to be present if the lowest or the highest score was achieved by more than 15% of the cases.

The Spearman’s test was adopted for correlation analyses. The criteria used for the interpretation of ICC and Spearman’s correlation coefficient were as follows: 0.00–0.25: no correlation; 0.26–0.49: low correlation; 0.50–0.69: moderate correlation; 0.70–0.89: high correlation; and 0.90–1.00: very high correlation [28].

Data were expressed in the text as median and 1st–3rd quartile or range (as indicated) and were reported in the tables as mean (±standard deviation). The threshold for statistical significance was set to *p* = 0.05. Statistical tests were performed with the IBM SPSS Statistics (version 20-IBM Corporation, Armonk, NY, USA) software package.

## 3. Results

### 3.1. Pain Intensity and Pain-Related Disability

No pain (NRS = 0) was reported by most of the patients (15 of 21 in resting conditions and 12 of 21 during movement). Median (1st–3rd quartile) values of NRS for pain were 0 (0–2) in both resting and movement conditions.

Median (1st–3rd quartile) values of Oswestry Disability Index score were 8 (6–22)%: pain-related disability was minimum for most of the patients (14 of 21 patients showed Oswestry Disability Index scores below 20%).

Surgical results could be considered excellent in most of the patients (all patients were working at the time of the surgery): consistently, 13 of 21 patients showed Majeed scores above 85%.

### 3.2. Physical Activity Estimates

The comparison of physical activity estimates between the two groups of subjects showed comparable (*p* = 0.65) values for patients (IPAQ score: 840 (420–1260) MET*min/week) and healthy controls (IPAQ score: 840 (420–900) MET*min/week). Consistently, the proportions of sedentary subjects (6 of 21 patients vs. 8 of 21 controls), moderately active subjects (13 of 21 patients vs. 12 of 21 controls), and active subjects (2 of 21 patients vs. 1 of 21 controls) were comparable (*p* > 0.05 for all comparisons) between the two groups.

### 3.3. Gait Parameters

At self-selected walking speed (Table 1), significant differences between patients and controls were observed for all spatiotemporal gait variables: walking speed, stride and step lengths, cadence, single support duration were lower in patients compared with controls, while double support duration and stance and swing phase durations were higher in patients compared with controls. Moreover, step length was lower and stance duration was higher for both limbs of the patients compared with controls.

At fast walking speed (Table 2), significant differences between patients and controls were observed for all but two (stride and step lengths) spatiotemporal gait variables: walking speed, cadence, and single support duration were lower in patients compared with controls, while double support duration and stance and swing phase durations were higher in patients compared with controls. Moreover, stance duration was higher for both limbs of the patients compared with controls.

Both groups of subjects were able to increase their walking speed when they were asked to walk at a faster-than-normal velocity, with no differences (*p* = 0.15) between patients (median (1st–3rd quartile) increase: +66 (47–77)%) and controls (median increase: +47 (37–66)%). Such an ability to increase the walking speed resulted from increases in cadence and stride length. The cadence increase was comparable (*p* = 0.97) between patients (+29 (16–32)%) and controls (+26 (17–38)%), while the stride length increase observed in patients (+22 (17–33)%) was significantly (*p* = 0.03) greater than that observed in controls (+18 (11–22)%).

### 3.4. Questionnaire Outcomes

No significant outcome differences were observed between the two administrations of both the Majeed (median (1st–3rd quartile)-1st administration: 85 (67–96) vs. 2nd administration: 92 (65–96); *p* = 0.18) and Iowa (1st administration: 90 (78–98) vs. 2nd administration: 92 (74–97); *p* = 0.53) questionnaires.

The ICC values were 0.95 (95% confidence limits: 0.88–0.98) and 0.94 (95% confidence limits: 0.86–0.96) for the Majeed and Iowa questionnaires, respectively. The SEM (SDC) values were 4.6 (12.8) and 3.7 (10.4) for the Majeed and Iowa questionnaires, respectively.

Significant negative correlations were observed both between the Majeed and Oswestry (*R* = −0.91, *p* < 0.0001) scores and between the Iowa and Oswestry (*R* = −0.83, *p* < 0.0001) scores, while a significant positive correlation was observed between the Majeed and Iowa (*R* = 0.81, *p* < 0.0001) scores. Significant positive correlations were also obtained both between Majeed score and walking speed (self-selected speed: *R* = 0.48, *p* = 0.03; fast speed: *R* = 0.59, *p* = 0.005) and between Iowa score and walking speed (self-selected speed: *R* = 0.48, *p* = 0.03; fast speed: R = 0.60, *p* = 0.004).

No floor effect was identified for the Majeed score (no cases scored the minimum value of 0 points), while a ceiling effect was observed (19% of the cases scored the maximum value of 100). No floor or ceiling effects were identified for the Iowa score (no cases scored the minimum value of 15 points and less than 15% of the cases scored the maximum value of 100 points).

## 4. Discussion

This study investigated pain intensity, perceived pain disability, physical activity, and spatiotemporal parameters of gait in patients after post-traumatic sacroiliac joint arthrodesis and tested the reliability and validity of the cross-culturally adapted Italian versions of the Majeed and Iowa questionnaires. The main findings of this study were that: (i) several years after surgery, most of the patients reported no pain and minimum pain-related disability; (ii) the physical activity profile was comparable between patients and healthy controls; (iii) patients showed an impaired walking performance compared with healthy controls; (iv) the Majeed and Iowa questionnaires showed excellent reliability, excellent (for the Majeed questionnaire) and good (for the Iowa questionnaire) construct validity, and poor convergent validity (for both questionnaires) relative to walking speed.

The use of a valid tool (gait analysis) for the assessment of walking performance and the recruitment of similar (i.e., matched for gender, age, and body mass index) populations of subjects are the notable strengths of this study. Another study strength is represented by the long-term follow-up. In fact, the experimental procedures were conducted after several years from surgery, at a time when muscle damage repair and bone healing had occurred, to minimize their influence on pain, physical activity and walking performance.

We found that several years after sacroiliac joint arthrodesis, most of the patients reported no pain and minimum pain-related disability and their physical activity profile was comparable with healthy controls. The excellent surgical results we observed (median values of the Majeed score ≥85 for both questionnaire administrations) are in agreement with the recent study by Baron et al. [7] who reported Majeed scores of 70 and 79, respectively, in pelvic-injured patients (with a minimum of three years of follow-up) treated with external and internal fixation. Such a positive surgical outcome in the long term may result from several factors: pre-surgical health status, severity of injury, detailed preoperative planning, surgical technique (the percutaneous fixation technique diminishes operative blood loss and operative time compared with the open reduction internal fixation) [2,29], excellent intraoperative fluoroscopic imaging (in cases treated with the percutaneous technique), accurate fracture reduction (which was excellent in most of our patients), and early patient mobilization [30].

However, the assessment of spatiotemporal parameters of gait showed that patients walked slower and used shorter steps compared with healthy controls. These walking pattern abnormalities have also been previously reported in patients after total knee [14] and hip arthroplasty [13]. The possible mechanism underlying these abnormalities include the post-intervention reduction of the hip and pelvis joint ranges of motion and the leg length discrepancy between the operated and non-operated side (which was not investigated). Although no direct recordings of central nervous system activity were obtained in this study, the observation that several between-group differences were documented for both limbs of the patients compared with controls suggests that the walking pattern abnormalities could also be underlain by neural determinants. From a clinical perspective, these findings highlight the relevance of (bilateral) re-education of affected neural pathways to supplement impaired neuromuscular performance after sacroiliac joint arthrodesis. On the other hand, all patients were able to increase their walking speed (as a result of the stride length and cadence increases) when they were asked to walk at a faster-than-normal velocity, similar to the increase observed in healthy controls. This ability to modulate the walking speed represents an important functional outcome, particularly in relation to pain and safety. In fact, failure to increase gait parameters has functional consequences, such as altered balance control and an increased risk of falls [13]. From a methodological perspective, an implication of the present results is that the assessment of walking parameters at both a self-selected comfortable and a fast velocity is recommended in orthopedic patients to investigate the walking performance modulation that could represent a clinically relevant surgical outcome measure.

Although the measurement properties (validity and responsiveness included) of the Majeed and Iowa scores have previously been investigated [9,31], to our knowledge this study is the first assessing the test-retest reliability and investigating SEM and SDC values of the questionnaire scores. Given the observed excellent reliability the Majeed and Iowa scores, another methodological implication resulting from this study is that these questionnaires, in combination with the assessment of spatiotemporal gait parameters, should be implemented in a routine clinical examination of surgical patients to properly design personalized rehabilitation programs.

There are several limitations of this study worth highlighting. First, the cross-sectional design does not allow causal inference. Second, physical activity was estimated through the IPAQ, which is a quick, simple, and inexpensive evaluation tool, although it is well-known that objective measures of physical activity demonstrate less variability in properties of methodological effectiveness than self-reported measures [32]. Third, we did not investigate the preoperative walking performance and physical activity of the patient group, although they represent a well-established predictor of post-operative performance and activity. Fourth, we did not perform specific investigations (e.g., range-of-motion assessment of the hip and pelvis joints, assessment of the leg length of the operated and non-operated side, muscle strength assessment, electromyography, and muscle ultrasonography) providing possible insights into the mechanisms underlying the walking impairment observed in patients. Fifth, we did not compare the algo-functional indexes and spatiotemporal parameters of gait between different subgroups of patients (i.e., patients treated with open surgery vs. percutaneous fixation) due to the smallness of the sample size.

## 5. Conclusions

This study documented long-term walking pattern abnormalities following sacroiliac joint arthrodesis despite excellent clinical results. The study also showed excellent reliability and construct validity of the Italian version of the Majeed and Iowa questionnaires. We propose that these questionnaires can be used in combination with the assessment of spatiotemporal gait parameters for the prognostic assessment and/or follow-up of surgical patients.

## Figures and Tables

**Table 1 jcm-09-02860-t001:** Spatiotemporal parameters of gait at self-selected speed. Mean (± standard deviation) values are reported. OL: operated limb. NOL: non-operated limb. C: controls.

PARAMETERS	PATIENTS		CONTROLS	*p* Value	
Walking speed (m/s)	1.0 (±0.2)		1.3 (±0.2)	0.001	
Stride length (cm)	121.4 (±16.2)		136.7 (±12.0)	0.002	
Cadence (steps/min)	102.2 (±10.6)		114.9 (±10.6)	0.001	
Double support duration (%)	23.2 (±4.3)		18.1 (±4.1)	0.001	
**PARAMETERS**	**PATIENTS**		**CONTROLS**	***p*** **Value**	**Post-Hoc Test**
	**Operated Limb**	**Non-Operated Limb**			
Step length (cm)	60.6 (±8.4)	60.6 (±7.9)	68.3 (±6.0)	0.002	OL vc C: 0.008 NOL vs. C: 0.006
Stance duration (ms)	729.5 (±84.7)	752.4 (±110.3)	631.9 (±78.5)	0.001	OL vs. C: 0.003 NOL vs. C: 0.001
Swing duration (ms)	467.4 (±55.5)	444.0 (±41.2)	430.6 (±27.1)	0.03	OL vs. C: 0.03
Single support duration (%)	37.5 (±3.6)	39.4 (±2.6)	40.9 (±2.1)	0.001	OL vs. C: 0.001

**Table 2 jcm-09-02860-t002:** Spatiotemporal parameters of gait at fast speed. Mean (± standard deviation) values are reported. OL: operated limb. NOL: non-operated limb. C: controls.

PARAMETERS	PATIENTS		CONTROLS	*p* Value	
Walking speed (m/s)	1.7 (±0.4)		2.0 (±0.3)	0.003	
Stride length (cm)	153.3 (±23.8)		160.6 (±12.3)	0.31	
Cadence (steps/min)	129.9 (±15.6)		146.9 (±15.6)	0.001	
Double support duration (%)	14.2 (±5.6)		10.5 (±3.4)	0.02	
**PARAMETERS**	**PATIENTS**		**CONTROLS**	***p*** **Value**	**Post-Hoc Test**
	**Operated Limb**	**Non-Operated Limb**			
Step length (cm)	76.3 (±11.5)	76.5 (±12.2)	80.1 (±6.0)	0.52	-
Stance duration (ms)	534.2 (±79.3)	547.0 (±109.5)	459.9 (±57.3)	0.001	OL vs. C: 0.006 NOL vs. C: 0.003
Swing duration (ms)	408.3 (±50.1)	387.0 (±31.4)	365.6 (±31.4)	0.004	OL vs. C: 0.003
Single support duration (%)	41.8 (±3.9)	43.9 (±3.2)	44.5 (±1.6)	0.01	OL vs. C: 0.012

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
