# Peer review of "Algo-Functional Indexes and Spatiotemporal Parameters of Gait after Sacroiliac Joint Arthrodesis"

_jcm, 2020, doi:10.3390/jcm9092860_

Round 1
Reviewer 1 Report
This cross-sectional study is purported 1) to evaluate the reliability and validity of the Italian version of the Majeed and Iowa questionnaires, and 2) to investigate walking abilities and algo-functional long-term surgical outcomes in patients who underwent a sacroiliac joint arthrodesis after a fracture. Both the methodological aspects and the arguments seem adequate. The English language and style are also good.
Some suggestions:
- the interpretation of spatiotemporal numerical parameters expressed in median and quartiles (see tables) is not very simple. Although I agree with the authors that it is the most correct way to report data that have no normal distribution, I suggest describing in the tables the numerical data in mean and standard deviation, even if the statistical analysis was done using non-parametric tests
- the difference in step length between the operated and non-operated side in fast speed gait (Table 2) seems to be considerable. Isn't it statistically significant?
- The statement “the walking pattern abnormalities are underlain by neural determinants..."may not be entirely correct. I believe that also a reduction of the ROM of hip and pelvis joints may result in the walking abnormalities you have seen in your patients, as well as in a different stride length in the two lower limbs at an increased speed (see above). I understand that, unfortunately, the ROM of the main articular fulcrums of the lower limbs have not been evaluated in your study. Please, comment on these arguments in the discussion section
- Finally, the evaluation of any differences between patients treated with open surgery or percutaneous fixation would have been very interesting. I invite you to also comment on this limitation of the study (due to the smallness of the sample and the cross-sectional nature of the study) as I believe that the reader of your article would be very interested in this issue.
Author Response
ANSWERS TO REVIEWER #1
REVIEWER
This cross-sectional study is purported 1) to evaluate the reliability and validity of the Italian version of the Majeed and Iowa questionnaires, and 2) to investigate walking abilities and algo-functional long-term surgical outcomes in patients who underwent a sacroiliac joint arthrodesis after a fracture. Both the methodological aspects and the arguments seem adequate. The English language and style are also good.
ANSWER
We thank the Reviewer for the appreciation of the work and for constructive criticisms.
Point-by-point replies to each comment follow.
REVIEWER
The interpretation of spatiotemporal numerical parameters expressed in median and quartiles (see tables) is not very simple. Although I agree with the authors that it is the most correct way to report data that have no normal distribution, I suggest describing in the tables the numerical data in mean and standard deviation, even if the statistical analysis was done using non-parametric tests
ANSWER
Median and quartile values have been changed (in both tables) with mean and standard deviation values, as requested.
REVIEWER
The difference in step length between the operated and non-operated side in fast speed gait (Table 2) seems to be considerable. Isn't it statistically significant?
ANSWER
We checked again the statistical analysis and we found no significant difference between the lengths (Kruskal Wallis test: P=0.52). The average values of the operated and non-operated side are exactly the same, while the median values reported in the previous version of the manuscript seemed different.
REVIEWER
The statement “the walking pattern abnormalities are underlain by neural determinants..."may not be entirely correct. I believe that also a reduction of the ROM of hip and pelvis joints may result in the walking abnormalities you have seen in your patients, as well as in a different stride length in the two lower limbs at an increased speed (see above). I understand that, unfortunately, the ROM of the main articular fulcrums of the lower limbs have not been evaluated in your study. Please, comment on these arguments in the discussion section
ANSWER
We thank the Reviewer for the suggestion to introduce a note of caution (line 290) in the statement relative to the possible neural determinants underlying the walking pattern abnormalities observed the patient group.
The fourth paragraph of the Discussion (lines 285-287) has been integrated with the alternative explanation proposed by the Reviewer and the final paragraph of the Discussion (line 316) has been integrated with the acknowledgement of the study limitation represented by the lack of assessment of the hip and pelvis joint ranges of motion.
REVIEWER
Finally, the evaluation of any differences between patients treated with open surgery or percutaneous fixation would have been very interesting. I invite you to also comment on this limitation of the study (due to the smallness of the sample and the cross-sectional nature of the study) as I believe that the reader of your article would be very interested in this issue.
ANSWER
We thank the Reviewer for this interesting comment. The final paragraph of the Discussion (lines 319-321) has been integrated with the acknowledgement of this study limitation.
Reviewer 2 Report
This paper evaluated the gait and functional results after SI joint arthrodesis.
- I wonder that authors translated the Majeed pelvic score and the Iowa pelvic score. Those questionaire included very simple and easy questions in English. It is questionable whether this validity study for translated version is necessary.
- Sample size calculation was done based on the questionaires. In order to compare the difference in gait parameters, authors should perform the sample size calculations base on gait parameters such as cadence or step length.
- The demographics of included participants such as follow-up period and previous injuries should be described on the manuscript.
- Authors commented on the displacement on fused SI joint. I wonder whether the leg length discrepancy in each patient is different. It is inappropriate to group patients with different leg lengths in the same group.
- It is too comprehensive comparison method to conclude that patients with fused SI joint have gait abnormality just because there is difference in step length or cadence between operated group and normal control. Step length and cadence can be different when tested twice in one person. If the authors want to investigate the gait abnormality in operated group, they have to check the pelvic anterior or posterior tilt, rotation et al in operated group.
Author Response
REVIEWER
This paper evaluated the gait and functional results after SI joint arthrodesis.
ANSWER
We thank the Reviewer for the appreciation of the work and for constructive criticisms.
Point-by-point replies to each comment follow.
REVIEWER
I wonder that authors translated the Majeed pelvic score and the Iowa pelvic score. Those questionaire included very simple and easy questions in English. It is questionable whether this validity study for translated version is necessary.
ANSWER
We agree with the Reviewer that the questions of both questionnaires are very simple. However, we think that the cross-cultural adaption process and the publication of a validated Italian version of both questionnaires can be useful to standardize their future use in both clinical and research applications. This is particularly relevant for the Majeed Pelvic Score that in the original publication by the Author proposed a range of possible scores (0-1 or 0-2 or 0-4 or 0-5) for the most severe level of dysfunction (i.e., the first category) of each item. As recently highlighted by Kleweno et al. (Inaccuracies in the Use of the Majeed Pelvic Outcome Score: A Systematic Literature Review. J Orthop Trauma. 2020;34:63-69) this implied inaccuracies in the use of the score and difficulties in the comparison between the scores reported by different studies. In the present Italian version of the Majeed questionnaire, the first category of each item was scored 0: therefore, we think the future use of our questionnaire version will enable the unbiased comparison between the scores reported by different (Italian) studies.
REVIEWER
Sample size calculation was done based on the questionnaires. In order to compare the difference in gait parameters, authors should perform the sample size calculations based on gait parameters such as cadence or step length.
ANSWER
This is an important issue that we have carefully considered. Power analysis was performed, according to previous studies, also on the basis of the walking velocity.
The following sentence has been added:
“Power analysis was also performed according to previous studies [13,14] and showed that a minimum sample size of 18 subjects in each group was required to detect significant walking velocity differences between patients and controls (a=0.05 and b=0.05)”.
REVIEWER
The demographics of included participants such as follow-up period and previous injuries should be described on the manuscript.
ANSWER
The demographics of included participants was reported in the paragraph 2.1, that has been integrated with the median (min-max range) duration of the follow-up (lines 72-74: “The present cross-sectional study was conducted after a median of 7 (min – max range: 5 – 10) years from surgery”).
The history of previous injuries was not available in the medical files and was not collected during the experimental procedures
REVIEWER
Authors commented on the displacement on fused SI joint. I wonder whether the leg length discrepancy in each patient is different. It is inappropriate to group patients with different leg lengths in the same group.
ANSWER
We thank the Reviewer for this comment.
The fourth paragraph of the Discussion (lines 286-287) has been integrated with the alternative explanation proposed by the Reviewer and the final paragraph of the Discussion (lines 316-317) has been integrated with the acknowledgement of this study limitation.
REVIEWER
It is too comprehensive comparison method to conclude that patients with fused SI joint have gait abnormality just because there is difference in step length or cadence between operated group and normal control. Step length and cadence can be different when tested twice in one person. If the authors want to investigate the gait abnormality in operated group, they have to check the pelvic anterior or posterior tilt, rotation et al in operated group.
ANSWER
We thank the Reviewer for this comment that was also provided by the other Reviewer.
The fourth paragraph of the Discussion (lines 285-287) has been integrated with the alternative explanation proposed by both Reviewers and the final paragraph of the Discussion (line 316) has been integrated with the acknowledgement of the study limitation represented by the lack of assessment of the hip and pelvis joint ranges of motion.
Round 2
Reviewer 2 Report
Authors revised the manuscript according to my comments.
This manuscript is a resubmission of an earlier submission. The following is a list of the peer review reports and author responses from that submission.